# Invited perspectives:
# How machine learning will change flood risk and impact assessment

Dennis Wagenaar[1,2], Alex Curran[1,3], Mariano Balbi[4], Alok Bhardwaj[5], Robert Soden[6,7,8] ,Emir Hartato[9], Gizem Mestav Sarica[10], Laddaporn Ruangpan[11], Giuseppe Molinario[7], David Lallemant[5,8]

5   1 Deltares, Delft, the Netherlands
2 VU University, Amsterdam, the Netherlands
3 Delft University of Technology, Delft, the Netherlands
4 University of Buenos Aires, Buenos Aires, Argentina
5 Earth Observatory of Singapore, Nanyang Technological University, Singapore
10  6 Columbia University, New York City, New York, United States of America
7 GFDRR, World Bank Group, Washington D.C., United States of America
8 Co-Risk Labs, Oakland, California, United States of America
Planet, San Francisco, United States of America
Institute of Catastrophe Risk Management, Nanyang Technological University, Singapore
11 IHE Delft Institute for water education, Delft, the Netherlands

*Correspondence to*: Dennis Wagenaar (dennis.wagenaar@deltares.nl)

**Abstract.** Increasing amounts of data, together with more computing power and better machine learning algorithms to analyse the data are causing changes in almost every aspect of our lives. This trend is expected to continue as more data keeps becoming available, computing power keeps improving and machine learning algorithms keep improving as well. Flood risk and impact
assessments are also being influenced by this trend, particularly in areas such as the development of mitigation measures, emergency response preparation, and flood recovery planning. Machine learning methods have the potential to improve accuracy as well as reduce calculating time and model development cost. It is expected that in the future more applications become feasible and many process models and traditional observation methods will be replaced by machine learning. Examples of this include the use of machine learning on remote sensing data to estimate exposure or on social media data to
improve flood response. Some improvements may require new data collection efforts, such as for the modelling of flood damages or defence failures. In other components, machine learning may not always be suitable or should be applied complementary to process models, for example in hydrodynamic applications. Overall, machine learning is likely to drastically improve future flood risk and impact assessments, but issues such as applicability, bias and ethics must be considered carefully to avoid misuse. This paper presents some of the current developments on the application of machine learning in this field and
highlights some key needs and challenges.

## 1.Introduction

Exponentially increasing computing power and data, as well as rapidly improving machine learning algorithms to analyse this data have been changing many aspects of our lives (Manyika et al., 2011).  These trends are expected to continue and will
undoubtedly keep affecting many scientific, commercial and social sectors (Manyika et al., 2011). Flood risk and impact assessments are no exception to this trend. Flooding yearly affects more people than any other natural hazard types (Jonkman, 2005) and the impact and frequency of flooding events is expected to increase in the future due to urban

development and climate change (Kundzewicz et al., 2014). It is therefore an opportunity for researchers and flood managers to tap into the potential of machine learning, taking advantage of their strengths while being cognisant of their limitations. It is also important to anticipate improvements in the capabilities of machine learning methods, so as to plan for forthcoming changes in flood modelling.

When assessing the interaction between floods and society, three different components can be recognized: exposure, hazard, and impact (Kron, 2002). Exposure refers to the characteristics of the people and assets that can be affected by flooding. Hazards are the physical characteristics of a flood such as the extent, water depth, duration and flow velocity. Impacts are the effects the hazard has on the exposure. To assess these three components, we make the distinction between flood risk, as the probabilistic analysis of the potential (predictive) impacts of floods and flood impact assessment, as the post-event assessment of (descriptive) impact from an actual flood event.[2] Table 1 provides examples of predictive and descriptive assessments in relation to the hazard, exposure and impact components. The scope of this paper is limited to the predictive and descriptive assessments shown in table 1 and doesn't include potential uses of machine learning in risk awareness or communication strategies.

Flood risk and impact assessments have many different applications. A useful paradigm to look at these different applications is the 'disaster management cycle' (Khan et al., 2008; National Research Council, 2006) (Fig 1). This cycle delineates the phases between events, i.e. the immediate response to an event, the long-term recovery, the mitigation to prevent future events and the preparation prior to a new forecasted event.

In the response phase, the focus is typically on descriptive hazard, exposure and impact assessments (e.g. Klemas, 2015), sometimes complemented with predictive models if the event descriptive information isn't available yet (e.g. a predictive model estimating the number of people affected can be fed by a descriptive hazard model of the flood extent). The challenge in this phase is mostly data reliability. In the recovery phase, descriptive assessments are often used for payouts (e.g. indemnity insurance), and one of the main challenges is ensuring these payouts are timely and reliable. In the mitigation phase, probabilistic predictive models are used (e.g. Wagenaar et al., 2019), typically for the design of risk-reduction interventions ranging from protective infrastructure to insurance products. The challenge in this phase is model reliability and uncertainties about future developments (e.g. uncertainty in future exposure). In the preparation phase, predictive models are used for emergency planning (e.g. Coughlan de Perez et al., 2016), where the challenge is the reliability, availability and communication of data. Machine learning is capable of generating more reliable and faster models that can help solve some of the current challenges in the disaster management cycle but could also provide new opportunities (GFDRR, 2018).

Machine learning algorithms can find patterns in data and use these patterns to make predictions about new data (Bishop, 2006). For example, when providing a machine learning algorithm with aerial images of either urban or rural areas and

corresponding labels (urban or rural) it can build the capacity to classify new unlabelled aerial images as either urban or rural. Features in the above example would be different components of the aerial images (i.e. pixel tone and locations) and the target variable would be the label (i.e. urban or rural). When a precise value is required as opposed to a label, it is called a 'regression task'(e.g. Bishop, 2006). An example of this is in flood damage modelling, where features such as water depth, flow velocity and building materials can be used to predict a target variable such as monetary economic damage based on historical records (e.g. Merz et al., 2013; Wagenaar et al., 2017). Due to the use of labelled training data (e.g. classified images or historic damage examples), regression and classification are called supervised learning tasks. Machine learning method categories also include unsupervised learning and reinforcement learning, (see GFDRR, 2018). However, such methods are not discussed in this paper because they are expected to have a smaller short-term impact on the field of flood risk and impact assessments.

The simplest machine learning algorithms have been used for a long time and are often known as basic statistical techniques (e.g. linear regression: Legendre, 1805; Gauss, 1809). More sophisticated machine learning techniques that emerged in the 1980s and 1990s (e.g. Decision Trees and Neural Networks) can find more complex non-linear patterns (Breimann et al., 1984; Rumelhart et al., 1986). Recent advances in machine learning (e.g. convolutional neural networks) make computer vision and other advanced applications possible (Krizhevsky et al., 2012). The more advanced techniques such as decision trees, neural networks and especially convolutional neural networks can find more complex patterns. This is because they allow for more complex non-linear functions to be fitted to the data. Such complex functions require a large number of model coefficients to be set during the training of the model. To set all these coefficients a lot of training examples are required. In some cases the number of training examples can be reduced with transfer learning techniques (Olivas et al., 2010). These techniques make it possible to re-use knowledge gained from other problems to train a model on a smaller training data set.

From the beginning, machine learning has been used in predictive flood hazard modelling (Solomatine & Ostfield, 2008) mostly as a faster and simpler alternative to process models. A simple example of this is the prediction of river discharge based on upstream rainfall data (e.g. Dibike & Solomatine, 2001). This type of modelling has been practiced for a long time but hasn't displaced the traditional process models. This is probably because the methods aren't sufficiently better than traditional methods to offset some disadvantages as discussed in the predictive hazard section. In recent years, more data is becoming available through remote sensing, social media (e.g. Fohringer et al., 2015), citizen science (e.g. Annis & Nardi, 2019) and other sources. This impulse of new data combined with machine algorithms could lead to changes in flood risk and impact assessment. Some of these changes have already been highlighted by major international organizations such as the World Bank and others (GFDRR, 2018).

This invited perspective paper starts with a perspective per risk assessment component as defined in table 1. These specific perspectives start with a description of the traditional approach for the assessments. Followed by a literature review on how machine learning techniques are currently being developed to improve the traditional approach and then proceed to speculate

on potential future improvements. This is followed by a general perspectives chapter in which general trends that come back in the different components are identified and discussed. This includes common challenges (i.e. data limitations, transferability, ethics and bias) and ends with some speculation about the likelihood of future developments.

## 2. Perspective per component

### 2.1 Exposure assessment

#### 2.1.1 Descriptive exposure assessments

Descriptive exposure assessments consist of detecting and characterizing (spatial) features such as current buildings and infrastructure, agriculture fields, roads and other infrastructure. Traditionally this has been done by population censuses, building counts and conventional mapping techniques that require ground surveys. Remote sensing is currently changing this. It has become common that aerial and satellite images are being ~~The manual~~manually digiti~~zed~~ and labelled ~~za~~t~~ed~~~~ion~~ ~~of~~ to make building footprints or map roads ~~using aerial or satellite images has become increasingly common~~. This ~~digitization of building footprints, but also notably of other features such as roads,~~ has been done ~~manually~~ by "crowds" of mappers in "mapathons", for example using the OpenStreetMap platform. Machine learning is very likely going to drastically change this. Research to automatically labelling remote sensing data has already been going on for some (e.g. Heermann & Khazenie, 1992; Giacinto & Roli, 2001). It is currently already being used to label build-up areas based on nighttime lights (Goldblatt et al., 2018) or satellite images (Goldblatt et al., 2016). Furthermore, algorithms are already being used to automatically label buildings (Sermanet et al., 2013; Alshehhi et al., 2017; GFDRR, 2018) and map roads (Gao et al., 2019) using aerial / satellite imagery. This will reduce the need for manual detection and will probably provide global availability of such building footprints and road information in the near future.

Part of an exposure assessment is the observation of asset features relevant for risk analysis. For example, building materials, building occupancy (e.g. residential or industrial), building height, ground floor elevation, poverty rates in the population, etc. This information is typically not available, but could be very valuable as input for impact models (e.g. Merz et al., 2014; Wagenaar et al., 2017; Schröter et al., 2014) or, for example, to account for poverty in cost-benefit analyses (e.g. Kind et al., 2016). Similarly, ground floor elevation information could radically improve urban pluvial flood damage modelling as damage from small-scale floods is very sensitive to such variables.

Some work has already been carried out on detecting poverty (Watmough et al., 2019) and building heights (Saadi & Bensaibi, 2014) by satellite imagery. Another source of this building feature information could be 360-degree street view images combined with computer vision techniques. Such images are available in, for example, the open source streetview data platform

Mapillary (Neuhold et al., 2017). Such techniques are already starting to impact earthquake risk assessments, such as in Guatemala, where 360 degree imagery was fed into Mapillary algorithms in order to automatically detect "soft story" buildings; those most likely to collapse in an earthquake. This was done by having the machine learning algorithm detect features that were indicators of large openings on the ground floor of buildings (large doors, garage doors, shop windows, etc.) (GFDRR, 2018). Computer vision techniques from street level imagery are currently limited to detecting such relatively simple features. However, based on recent advances seen in other computer vision applications (e.g. facial recognition), it is likely that in the future it will be possible to detect more complex building features also. For computer vision models to detect complex information like ground floor elevation or building materials, it would be necessary to provide labelled examples to the algorithms. Such labelled examples are sometimes already available for some areas, e.g. ground floor elevation (Bouwer et al., 2017) or building materials (Schröter et al., 2018).

### 2.1.2 Predictive exposure assessments

Predictive exposure mapping consists of estimates of future exposure. This mostly includes modelling to predict urban growth and other changes in land-use. It is required for evaluating flood mitigation measures (e.g. Wagenaar et al., 2019) because such measures typically need to function for a long time and should therefore still perform as required after predicted land-use changes. Land-use changes affect the impact of a flood because more damage may occur for the same flood hazard and the flood hazard may become greater because of changes in impervious area and therefore rainfall-runoff (Triantakonstantis and Mountrakis, 2013; Mestav Sarica et al., 2019). Predictive exposure assessments for flood risk and impact assessments are currently often not carried out spatially, but rather GDP growth projections are applied to estimate future total exposure values (e.g. van der Most et al., 2014; Wagenaar et al., 2019). This is enough for some studies but if large land-use changes are expected a land-use change or urban growth model is required.

Urban growth has been modelled with simple machine learning models in the past (e.g. logistic regression) (Samardzic-Petrovic, 2017). The use of Cellular Automata (CA) models has become more common recently (Naghibi et al., 2016). These models assign cells as either urban or non-urban based on specific transition rules. Determining the optimum transition rules is a critical issue for CA modelling (Aarthi and Gnanappazham, 2019). This is sometimes difficult because of human bias, heterogeneity and nonlinear relations between driving factors and urban expansion (Naghibi et al., 2016; Xu et al., 2019). To overcome these limitations, machine learning algorithms such as artificial neural networks have been integrated with traditional CA to model urban growth (Aarthi and Gnanappazham, 2019; Naghibi et al., 2016). They then use historical land-use changes (e.g. Song et al., 2015) to learn the transition rules. Complex machine learning models have also been directly applied to urban growth modelling without the CA model structure (Pal and Ghosh, 2017). These improvements, together with more data about past land-use changes and additional computation power, are expected to provide better future land-use maps and make high-resolution future land-use maps globally available.

## 2.2 Hazards assessment

### 2.2.1 Descriptive hazard assessments

Descriptive flood hazard assessment focuses primarily on the response phase, i.e. in estimating current inundation extents and depths to assist both emergency responders and those affected directly. This is traditionally achieved using optical remote sensing data, local sensors or manually collected data from observers on the ground. However, the rise of two major data sources, Synthetic Aperture Radar (SAR) and social media, provides a number of opportunities for machine learning to improve upon current flood detection methods.

During a flood event, affected populations frequently produce 'user-generated content' or 'crowd-sourced' data from social media posts or apps where citizens can report floods (Mazoleni et al., 2017; Assumpção et al., 2018; Annis & Nardi, 2019; UrbanRiskLab, 2019). This is especially the case in urban areas where internet and social media penetration are higher compared to rural areas. This data is often 'tagged' temporally and spatially and can be used by machine learning algorithms for applications such as nowcasting by searching for certain keywords like "flood" (e.g. see Tkachenko et al. 2017, Bischke et al. 2017, Lopez-Fuentes et al. 2017). The method is currently used to map real-time flood extents in several countries (Eilander et al., 2016). Potential future machine learning and computer vision techniques could be extended to estimate water depths and other flood characteristics from posted photos.

Remotely-sensed optical data is often used to identify the extents of flooding, but optical sensors are not functional during periods of cloud-cover or at night. Furthermore, the temporal resolution often prevents the observation of flash floods. SAR data using the microwave wavelengths of the electro-magnetic spectrum can help overcome these problems by providing additional imagery during the night or during cloud cover. Adding night-time and cloud-cover images will provide a higher total temporal resolution. Flood extents are currently determined with statistical methods using thresholds to subsequently identify flood extents e.g. by using Bayesian method on SAR amplitude time-series data (Lin et al. 2019). Advanced machine learning classification methods are being developed to improve this process, but in order to train them it is necessary to have manually labelled images as training data. Collection of this labelled flood extent information is the main challenge for automatic detection moving forward. Manual methods could harness the power of the crowd, as people are connected through the internet or with mapathons. These approaches could have game-changing implications for the training of machine learning algorithms. Already mapathons are often 'trainathons', where mappers are not only manual digitizers, but also labellers and trainers for automated machine learning methods for the future.

### 2.2.2 Predictive hazards assessments

Predictive flood hazard assessments consist of predicting future floods and their characteristics such as extents, inundation depths, durations, flow velocities, waves and water levels in rivers or seas. These assessments are applied for short-term forecasting in the preparation phase (preparing for imminent events) and long-term risk analyses for use in flood risk management (mitigation phase).

In flood-forecasting, traditional methods of predicting hazard variables can involve a chain of hydrologic and hydraulic models that describe the physical processes. Although such models provide system understanding, they often have high computational and data requirements. Therefore, the use of process models may not always be feasible or necessary in the preparation stage of a disaster. At that moment, accurate and timely outputs become more important than system understanding, and the use of 'black-box' machine learning models (e.g. Campolo et al., 2009) is becoming more widespread (Mosavi et al., 2018). The increased speed can create a trade-off with the robustness of forecast models, as changes to the hydraulic system (such as a new structure that could be easily implemented into a hydraulic model) cannot be directly introduced into a trained machine learning model. In addition, machine learning models might not perform well in predicting extremes far outside past observations, since it has not been trained against such extremes.

A review of flood forecasting methods using machine learning by Mosavi et al. (2018) highlights trends such as component and ensemble models (collectively termed 'hybrid models', Corzo & Solomatine, 2014). Hybrid component models assign machine learning a specific task in the modelling process that is either highly complex or not well understood. Examples of this include using machine learning for error correctors (see, for example, studies by Abrahart et al. 2007 and Google Research - Nevo et al. 2019) or flows subject to human influence (Yaseen et al., 2019). Hybrid ensemble methods often use machine learning models to supplement process models, providing robust predictions and uncertainty ranges (Solomatine & Ostfeld, 2007). Such methods benefit from the speed and ability to deal with non-linear multi-variable problems of machine learning modelling and the process understanding available in conventional modelling. The review by Mosavi et al. (2018) does not consider gridded / spatial forecasting techniques, but advanced machine learning techniques are starting to be developed for precipitation pattern nowcasting (Shi et al. 2015) and flood extents prediction (Chang et al. 2018). Another application of machine learning in the preparation phase is in the real-time control of flood defences and systems (e.g. Lobbrecht & Solomatine, 2002; Castelletti et al., 2010). For example, Lobbrecht & Solomatine (2002) used machine learning methods to optimise control decisions in the event of communication network breakdowns during extreme storm events.

Another major application for machine learning in long-term risk analysis is 'surrogate' modelling (Ong et al. 2008), in which the outputs from process models are used to train computationally less-intensive machine learning models. This can be applied to speed up different types of process models applied in predictive hazard modelling. For example, in flood defence analysis

and design, classical reliability techniques such as First Order Reliability Methods (FORM) and Monte-Carlo simulations (Steenbergen et al. 2004), or large-scale risk analyses that utilise them (Curran et al. 2019), can be replicated using a relatively small amount of evaluations as samples (Chojaczyk et al. 2015, Kingston et al. 2011). However, surrogate models may be particularly susceptible to extrapolation problems, where input data outside the range of the training data is introduced (Ghalkhani et al., 2013).

In the mitigation phase a chain of hydrologic and hydraulic models that describe the physical processes is typically applied (e.g. Wagenaar et al., 2019). In general, system understanding is required to assess proposed or potential future changes. In such cases, data-driven approaches are typically not applicable as there is no data about how the system behaves after the changes occur and hence simulation models are required that describe the physical system.

## 2.3 Flood impact assessment

### 2.3.1 Descriptive impact assessments

Descriptive impact assessments consist of making estimates of the flood impact after or during an event. This is traditionally done with manually collected data from observers on the ground. However, such manual ground inspections are slow and require people to enter the disaster area. Remote sensing can be used to get a very quick first impression of the damage to help with disaster response. Such techniques have already been applied, for earthquake and wind damage (e.g. Menderes et al., 2016). For flooding, this is often more difficult because damage inside buildings is difficult to obtain either from aerial-based or space-based sensors. Only when buildings completely collapse or are removed by strong flows does remote sensing become feasible. This is, for example, the case with flash floods, tsunamis or some storm surges. 360-degree streetview images collected after a flood could potentially be used for damage assessment. Machine learning techniques could then eventually be used to give a quick first estimate of the damage.

The use of machine learning techniques for automatic detection of damages from remote sensing information (aerial or streetview) requires labelled training data from manually collected data from observers on the ground. This data is currently rare. An approach could be to start using remote sensing data to manually label the impact. A way to get around this limitation is to detect changes in pre- and post-flood using high-resolution satellite images for urban areas where many buildings are damaged. Pixels with changed information will denote the damage that happened due to the floods. Eventually this data can then be used as training data for cases where only the post-flood images are available within a short time interval after the flood event. This method would however only be relevant for catastrophic floods because it doesn't address the fact that most damage remains not observable from top-view. On top of that this approach introduces significant new error: (1) error in the change detection signal, (2) error in relating the change to damage, (3) error in training a new model based on those damage

labels. Imagery from different angles (e.g. from streetview or drones) might be more useful for change detection, however this data would also be more difficult to acquire.

### 2.3.2 Predictive impact assessments

Predictive flood impact assessments include models that translate hazard and exposure information into socio-economic
impacts of the flood. This can include information such as      monetary flood damage, casualties, buildings damaged, crop damage, disease outbreak, building materials needed, recovery time, health monitoring of key structures or indirect damage (damages that occur in a different spatial and/or temporal setting than the originating event).

Most predictive flood damage modelling relies on depth-damage functions that describe a relationship between the water depth
and monetary flood damage (Merz et al., 2010). They are either based on historical flood damage records (e.g. Thieken et al., 2008; Kreibich et al., 2010) or on expert estimates (e.g. Penning-Rowsell et al., 2005). In practice, many more variables than water depth have an influence on the flood damage (Cammerer et al., 2013, Wagenaar et al., 2016). Therefore, in the scientific literature there has been a move towards multi-variable flood damage models that use many variables (e.g. flood duration, velocity, building materials, socio-economic status of inhabitants etc.) instead of just water depth (e.g. Merz et al., 2013;
Spekkers et al., 2014 Chinh, 2015; Kreibich et al., 2017; Wagenaar et al., 2017; Carisi et al., 2018; Amadio et al., 2019). These models are based on data and machine learning. The problem lies with insufficient data availability to train machine learning models and that using the models requires a lot of feature data about flood and building characteristics plus socio-economic data about inhabitants (Wagenaar et al., 2017). In the future we expect more data about features to become available from computer vision applied to street view, satellite or drone images (see descriptive exposure section). This would improve the
quality of such models, could make it easier to apply them and make the development possible for more areas.

Machine learning could also be applied to predict disease outbreak after floods by combining remote sensing, meteorological, and socio-economic data (e.g. Mayfield et al., 2018; Carvajal et al., 2018; Modu et al., 2017; Yomwan et al., 2015; Tiwari et al., 2013; Shively et al., 2015). In a flood event, there is an increased risk of infectious diseases among survivors and displaced
persons such as measles, diarrhea, acute respiratory infections and malaria can be responsible for many deaths (Lignon, 2006). Predictive modelling of such diseases is rarely carried out, and current approaches mostly focus on simple regression models or process models that simulate the spread of pollutants in the water. One major challenge is that the degree to which such epidemics occur, is associated with the regional endemicity of specific diseases, the nature and scope of the disaster, the level of public health infrastructure in place both before and after the event, and the level and efficacy of disaster response (Ivers &
Ryan, 2006). Machine learning models could take such complex processes better into account.

Machine learning can be used for structural health monitoring, this has applications in the preparation phase (Pyayt et al. 2014, Jonkman et al. 2018) and in the long-term reliability analysis required in the mitigation phase (Prendergast et al. 2018, Klerk

et al. 2019). Structural health monitoring is in the preparation phase is often done by manual inspections of the infrastructure on the ground, for example in the Netherlands there is a large network of volunteers that can be activated in case of high river levels to inspect the dikes. In the mitigation phase this is done by geotechnical process models fed by observations from the ground (e.g. De Waal, 2016), this is for example applied to decide on dike strengthening. Machine learning algorithms can help detect damage patterns from sensor data and are currently being used for the monitoring of flood defence structures such as dikes (Pyayt et al. 2011). Similar methods have also been applied to bridges (Neves et al. 2017). The use of both machine learning algorithms and traditional techniques for damage detection during floods is still very scarce (Prendergast et al., 2018, Pyayt et al., 2011); however, integration of structural health monitoring with flood early warning systems is a very promising field of development for machine learning techniques but would also requiring training data.

Indirect damages and business interruption are often taken into account simply through a scaling factor of the direct damage (e.g. Wagenaar et al., 2019). More complex models to quantify such damages include input-output models and general equilibrium models (e.g. Koks et al., 2016). To quantify indirect damages, such as business interruption losses, estimating the time it will take for different assets to be back in full or partial functionality is required. These post-disaster restoration models have started to be formalized in the last few years, primarily focused on earthquake disasters (Kang et al., 2018; Burton et al., 2018). Due to a lack of gathered empirical data on post-disaster recovery, the use of data-intensive machine learning techniques has not yet made an impact on this discipline. However, the need of probabilistically quantifying recovery will require the use of statistical models for calibration or assessments of recovery times, and that might be possible in the near future with the use of new remote sensing and crowd-sourcing technologies to obtain the empirical feature data needed.

## 3. General Perspectives

### 3.1 Data limitations

Many machine learning applications in flood risk and impact modelling appear to be limited by a lack of data, especially training data needed to build effective machine learning models. This is especially true since the field of flood risk analysis is concerned primarily with extreme events, which are rare, and data-collection during such events is often logistically difficult. The increase in the amount of data around the world does not necessarily imply that this problem will be resolved in the future. Some data is simply not collected or there are measurement definition or quality issues. To fulfil the potential of machine learning, new data collection efforts will be required, along with data standardization protocols. This will take collaboration between different organisations and stakeholders, setting of data standards and a willingness to share. This problem is common to impact data collected (see 2.3.1 and 2.3.2), labelled flood extent data (see 2.2.1), social media hazard data (see 2.2.1) and first floor elevation data (see 2.1.1).

**3.2 Transferability of data**

A critical assumption behind machine learning techniques is that the data being used to train a model is representative of the situation the model needs to be applied in. For example, a dataset on damage to concrete buildings is not fully applicable to modelling the damage to thatched huts. It is therefore important to collect heterogenous datasets that cover a large spectrum of potential situations (Wagenaar et al., 2018). Data that isn't fully applicable can still have some value, for example through domain adaptation or transfer learning (GFDRR, 2018) but applicable data is always required as well. Wagenaar et al. (2020)

showed that sample selection bias correction, a form of domain adaptation, helps to improve machine learning impact models in a transfer setting.  Furthermore, it is important to work on efficient ways to communicate the applicability of data-driven models.

**3.3 Ethics and Bias**

Significant attention is currently being given to questions of the ethics and bias of machine-learning systems across a variety

of domains, including facial recognition (Keyes, 2018), automated weaponry (Suchman et al., 2016), criminal justice (Eubanks, 2018) and search engines (Noble 2018). A number of technology companies and research institutions have developed guidelines for evaluating machine-learning systems, but this work is still evolving. Despite similar potential for negative impacts of these tools in flood risk management (Soden et al., 2019), the community has not given these issues as much attention. Such concerns include the potential for reinforcing existing social inequalities and the reduced role of human

judgement in modelling processes. These are risks that need to be weighed seriously against the potential benefits of machine learning and explored in greater detail

Biases in machine learning can occur because of datasets that, for a number of reasons, do not fully represent the phenomena which they are meant to describe (e.g. people are accidentally excluded). For example, we often measure what we have data

for, rather than measure what matters most, or use training datasets that reinforce past problems. For example, if certain settlements aren't detected in exposure maps, because they use different construction practices than the settlements used in training datasets, they may not receive emergency aid in the event of a flood. These problems can be mitigated by ensuring modelers understand the context of what they are attempting to model. Other ethical issues raised by machine learning in the flood management context  include data ownership, transparency, consent, and privacy. For example, some people may object

to having their home labelled "vulnerable" on a vulnerability map used by first responders. Privacy concerns may be aggravated by machine learning and other big data techniques. Ethics problems should be addressed by carefully weighing the benefits of collecting certain data against the related privacy costs, in collaboration with people who may be affected by the outcomes of decisions based on machine-learning tools.

An additional ethical concern regarding machine learning in flood risk assessment is misuse of models. In some sectors great advances have been made with machine learning (e.g. facial recognition, self-driving cars). This success for some tasks, can lead to an awe-inspiring general attitude towards the techniques (Ames, 2018; Narayanan, 2019). This hype sometimes leads to unwarranted trust in the techniques for tasks machine learning is not (yet) suitable for (Narayanan, 2019). For example, many companies are currently using machine learning for hiring decisions despite well-documented failings of these tools.

(Narayanan, 2019; Raghavan et al., 2019). In order to avoid such misuse in flood risk assessment, it is important that machine learning implementations are transparent and supervised by independent machine learning and flood risk assessment experts.

   Importantly, flood risk assessments are highly data reliant, and the increased attention to questions of ethics and bias in machine learning systems might serve as an opportunity to drive conversations in our field about the limits of disaster data more broadly.

Many of the sources of bias or ethical concerns in machine-learning systems originate in, or share common roots with, other kinds of data used to understand disaster risks . This includes issues such as 1) property values driving what areas gets protected, 2) privacy concerns (which may be aggravated by ML and other big data techniques), 3) how the lack of gender/age/ethnicity disaggregated data on disaster risk masks differential vulnerabilities, and 4) the importance of public participation and the voice of residents of areas portrayed by models as "at risk". Detailed analyses of specific cases (e.g. Soden

& Kauffman, 2019) are urgently needed to make further progress in understanding the consequences of the assessment methods we use to understand disasters.

### 3.4 Future predictions

In the following section we draw some general conclusions about how machine learning will change flood risk and impact assessments. Table 2 provides an overview of these predictions.

**3.4.1 Very likely changes**

A few of the trends seem inevitable, primarily in cases where recent technological advances or data that recently became available make next steps obvious. A good example of this is the automatic detection of building footprints and roads from high resolution remote sensing imagery (see 2.1.1). This is already possible and will, especially in data-poor areas, drastically improve the quality of the first response and risk calculation. Further advances in the use of machine learning in descriptive

hazard assessment through social media are also inevitable (see 2.2.1), given the amount of data available to social media companies.

### 3.4.2 Likely and potential changes

This is the category that can be shaped the most by individual innovators and the majority of the advances discussed in this

paper fall under this category. In this case, the innovation still experiences some kind of obstacle that prevents widespread

application. It is typically difficult to predict whether such obstacles can be truly removed in the future and how long that will take. Because flood risk and impact assessments are a relatively small field, the obstacles are often economic feasibility that is difficult to assess combined with conservative users. An example of this is the large-scale collection of impact data which is required for both descriptive and predictive impact modelling (see 2.3.1 and 2.3.3) or the training data required for descriptive hazard assessments (see 2.2.1). Sometimes the obstacle is also technical feasibility, for example whether it will really be possible to extract first floor elevation levels from streetview (see 2.1.2). Innovations are also interdependent, for example, when building feature information can be automatically extracted from streetview, impact models will become easier to train and easier to run and it will make more sense to start collecting the required impact data.

### 3.4.3 Unlikely changes

For some processes, machine learning may not be the best solution from a theoretical perspective. For example, the processes of how water flows are very well known and can be well approximated with existing equations. It, therefore, does not always make sense to pick a machine learning approach. Another situation when machine learning is not applicable is when a system is being modelled on which predictions need to be made that cannot have been seen in the data or when we know from an exploratory data analysis that we have no data for it (GFDRR, 2018). For example, how a system may behave under never seen discharges or after new infrastructure has been built (e.g. new dam in the river). In these cases, machine learning may play a role in some components of the model, but process models will very likely remain crucial in simulating the never before seen conditions. Especially for predictive hazard models (see 2.2.2), there are many elements that are unlikely to change with the advance of machine learning.

### 3.4.4 New practices in flood risk and impact assessments

Most change to flood risk and impact assessments discussed in this manuscript relate to better models. Such cheaper, faster and more accurate models could possibly yield new practices in flood risk and impact assessments. Cheaper models would make flood risk and impact assessments feasible to carry out for a larger group of users and are therefore likely to make emergency aid and investments in mitigation measures more efficient. Faster methods may speed up emergency response and recovery, especially when manually collected data from observers on the ground are replaced by remote earth observation. More accurate models may lead to more early actions being feasible (Coughlan de Perez et al. 2014) and hence early actions can be carried out that couldn't be carried out before. For example, more targeted measures during the preparation and response phase of a flood. Such new measures include providing emergency payouts even before the event to the most vulnerable people (e.g. Reuters, 2019), prioritization of emergency measures in buildings, targeted disease outbreak prevention (Coughlan de Perez et al. 2014), early shipping of the right emergency goods (Coughlan de Perez et al. 2014)  and prioritization of early harvesting of crops.

## Acknowledgement

This invited perspective paper benefited from the unique intellectual environment created and facilitated through the Understanding Risk Field lab on urban flooding in Chiang Mai, Thailand in June 2019 (https://urfieldlab.com/). We would like to thank the organizers of the event (Robert Soden, David Lallemant, Perrine Hamel, Katherine Barnes, and Giuseppe Molinario) for providing the unique setting to make this paper possible. We would also like to thank the other participants who provided valuable input on this paper by participating in some of the discussions, namely: Gautam Dadhich, Carmen Acosta, Maricar Rabonza, Pamela Cajilig, Rahul Sharma and Wahaj Habib. Furthermore, we would like to thank the editor (Dr. Heidi Kreibich) and the reviewers (Dr. Fernando Nardi and an anonymous reviewer) for their contributions to the paper. This project is supported in part by the National Research Foundation, Prime Minister's Office, Singapore under the NRF-NRFF2018-06 award.

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

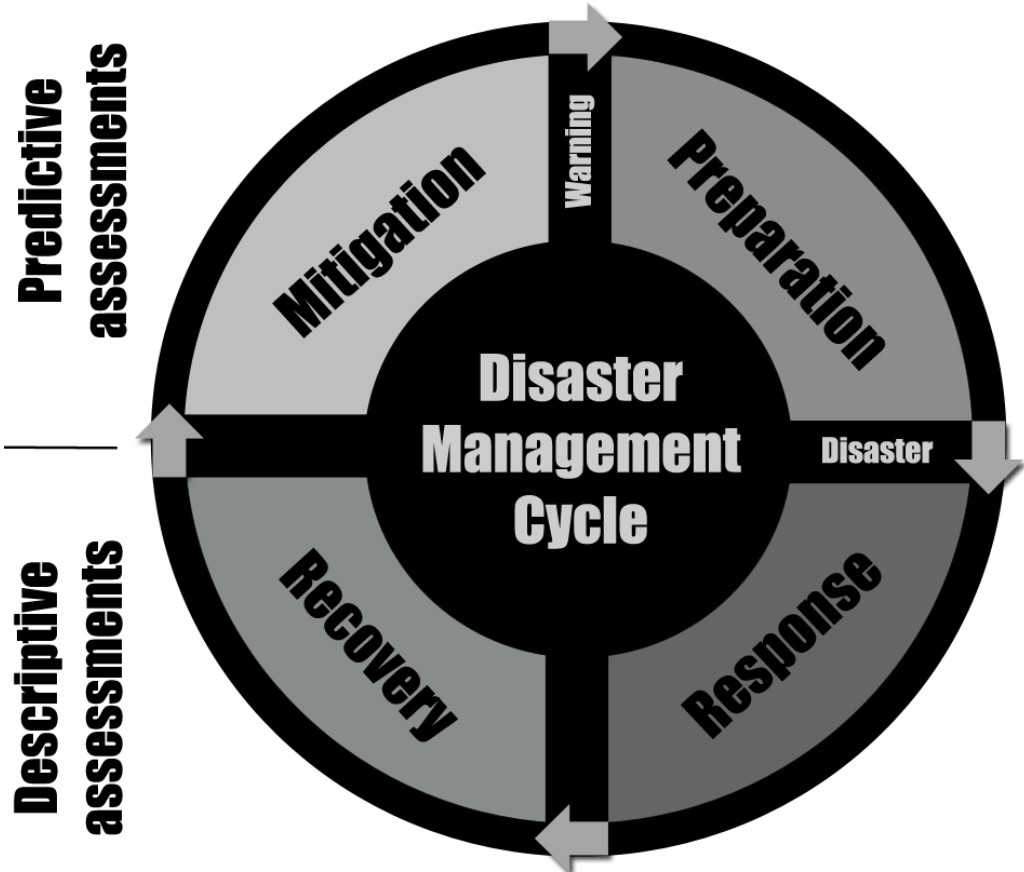

**Figure 1: Disaster management cycle, a common paradigm tool.**




**Table 1: Overview of different types of flood risk and impact assessments**

|  | Predictive | Descriptive |
|---|---|---|
| **Exposure** | Urban growth modelling | Identification of current built-up area |
| **Hazard** | Flood modelling | Mapping current and past floods |
| **Impact** | Forecasting impact (e.g. damage) | Assessing flood impacts (e.g. damage) after they have occurred |





**Table 2: Future predictions**

|  | **Predictive** | **Descriptive** |
|---|---|---|
| **Exposure** | Likely incremental changes, e.g improved Cellular Automata transition rules | Very likely significant changes e.g. automatic exposure detection including building features |
| **Hazard** | Diverse field, changes are more likely to be complementary or to specific components of modelling | Likely changes in detection due to remote sensing and social media algorithms. |
| **Impact** | Potential for significant changes (i.e. multi-variable data-driven methods) | Significant changes likely for some elements others will probably remain the same |
