# Peer review of "Invited perspectives: How machine learning will change flood risk and impact assessment"

_Natural Hazards and Earth System Sciences, 2019_

## Referee Comment (RC1) · Anonymous Referee #1 · 26 Nov 2019

Invited perspectives: How machine learning will change flood risk and impact assessment, by Wagenaar et al. – submitted to NHESS

Overall evaluation:

The manuscript reflects the result of the effort of a 2 weeks long collaboration during the Understanding Risk Field lab on urban flooding in Chiang Mai, Thailand, in June 2019 and intends give an overview of the principal developments and applications of machine learning methods in the flood risk field. It analyses the current and future role of machine learning for each component of the flood risk, taking also into account ethics and bias of this topic. The context is in line with the scope of the Journal and the aim of the manuscript is very interesting and useful for all scientist (and not only), who works on flood risk and are quite new with machine learning approaches, but want to

discover their potentialities and applicability contexts. I surely consider the manuscript suitable for publication, although I think it needs be even more improved by means of some minor revisions, that I'm reporting hereafter.

Minor comments:

Terminology: as general assumption of our community (e. g. Merz et al., 2010, de Moel et al., 200), and following what authors write at P2 L43-46, flood impact is one of the three components of flood risk. Therefore, I would avoid to use the statement "flood risk and impact assessment", because flood impact is somehow included in flood risk. Please, go through the manuscript (include title) and correct these cases.

Introduction: I would add some more detailed explanations of the different machine learning methods. Just some sentences, but it can help in order to have clear in mind, in the rest of the manuscript, what decision trees, neural network, etc. are. In addition, as general comments, I would add some sentences (and references) which state that machine learning methods can really improve estimations, in case of large datasets available: up to now, this concept is taken for granted, but citing some studies that demonstrate it could improve the manuscript, in my opinion.

Structure: in order to be consistent with the definition of flood risk at P2 L43-49, I would suggest to analyse hazard (current Ch. 3) before exposure (current Ch. 2).

Ch. 5: I would add, as an issue, the difficulty to use machine learning methods: they require a quite high degree of knowledges in order to really appreciate improvements in flood risk estimation and to avoid errors, that can easily be done by not-experts.

P3 L89-90: Why traditional process models were not displaced by machine learning methods? Please explain, or refer to specific following Sections.

P11 L352-353: I would suggest to remove the sentence.

P12 L383-384: please add reference for this statement.

Table 2 is never cited in the text. Please correct.

As suggestion, I list some recent papers I found, which use machine learning methods in the flood risk assessment, and can be useful to cite in order to strengthen some concepts: - Amadio, Mattia; Scorzini, Anna Rita; Carisi, Francesca; Essenfelder, Arthur H.; Domeneghetti, Alessio; Mysiak, Jaroslav; Castellarin, Attilio (2019): Testing empirical and synthetic flood damage models: the case of Italy. In Nat. Hazards Earth Syst. Sci. 19 (3), pp. 661–678. DOI: 10.5194/nhess-19-661-2019. - Campolo, M., Soldati, A., and Andreussi, P.: Artificial neural network approach to flood forecasting in the River Arno, Hydrolog. Sci. J., 48, 381–398, https://doi.org/10.1623/hysj.48.3.381.45286, 2003. - Carisi, F., Schröter, K., Domeneghetti, A., Kreibich, H., and Castellarin, A.: Development and assessment of uni- and multivariable flood loss models for Emilia-Romagna (Italy), Nat. Hazards Earth Syst. Sci., 18, 2057–2079, https://doi.org/10.5194/nhess-18-2057-2018, 2018. - Chinh, D., Gain, A., Dung, N., Haase, D., and Kreibich, H.: Multi-Variate Analyses of Flood Loss in Can Tho City, Mekong Delta, Water, 8, 6, https://doi.org/10.3390/w8010006, 2015. - Giacinto, G. and Roli, F.: Design of effective neural network ensembles for image classification purposes, Image Vis. Comput., 19, 699–707, https://doi.org/10.1016/S0262-8856(01)00045-2, 2001. - Heermann, P. D. and Khazenie, N.: Classification of multispectral remote sensing data using a back-propagation neural network, IEEE T. Geosci. Remote., 30, 81–88, 1992. - Kreibich, H., Botto, A., Merz, B., and Schröter, K.: Probabilistic, Multivariable Flood Loss Modeling on the Mesoscale with BT-FLEMO, Risk Anal., 37, 774–787, https://doi.org/10.1111/risa.12650, 2017. - Spekkers, M. H., Kok, M., Clemens, F. H. L. R., and ten Veldhuis, J. A. E.: Decision-tree analysis of factors influencing rainfall related building structure and content damage, Nat. Hazards Earth Syst. Sci., 14, 2531–2547, https://doi.org/10.5194/nhess-14-2531-2014, 2014. - Wang, Z., Lai, C., Chen, X., Yang, B., Zhao, S., and Bai, X.: Flood hazard risk assessment model based on random forest, J. Hydrol., 527, 1130–1141, https://doi.org/10.1016/j.jhydrol.2015.06.008, 2015.

---

## Referee Comment (RC2) · Anonymous Referee #2 · 3 Jan 2020

Foreword This invited perspective paper provides views and perceptions on a topical aspect of flood risk management, thus, a critical subject of interest for this journal. Authors provide a critical analysis on the upcoming scientific and practical breakthrough brought by machine learning (ML) to flood risk and impact assessment studies. The paper is structured into an introductory section and a second core part (referring to sections 2, 3 and 4) where the three main core concepts - exposure, hazard and impact – are analyzed with specific focus on descriptive versus predictive assessments related to flood exposure, hazard and impact studies. Authors discuss in the second core part how to date flood exposure/hazard/impact knowledge frameworks and predictions are developed as respect to how they will be made in the near future thanks to ML in particular. Afterwards, in section 5, the perspective part of the manuscript is provided with

[Figure]

Common challenges (Transferability of data, Ethics and Bias) and Future predictions (categorized upon the potential of the changes to happen). For example authors conclude that "automatic detection of building footprints" is very likely to change/happen soon supported by ML, while "unlikely changes" due to ML include the ML-supported development of better numerical physically based models of flood dynamics. Authors has surely knowledgeable and experienced on the topic, but the manuscript seems to miss to be adequately structured and supported by solid reasoning's, references and results. As a result, I'd suggest to improve this paper with a more in-depth description, critical analysis and discussion of the main conclusion of this work before publication. I'm sure authors have much more to give to the readership as respect to this first submission that seems to only scratch the surface. I'm also concerned by the lack of sound arguments and more extended referencing of published research studies to support the main findings, that are only somehow supported by general reasoning and insightful comments that has to be captured behind the lines. A more explicit description of outcomes from a larger number of key reference papers shall be integrated to better support this work, as requested to perspective papers, especially considering the case of ML for flood risk management has already matured in recent years with hundred of papers in the specific topic. Along the line of what I just stated in the foreword, I'll try to summarize in the following general and specific comments/remarks the points that authors should address to improve the paper to achieve the expected goals of this invited perspective.

General comments 1) Lack of adequate referencing. Key references of this work are the following: a. Bishop, C. M. (2006), Pattern Recognition and Machine Learning, Springer, ISBN 978-0-387-31073-2 b. The two works: Solomatine, D.P., Ostfield, A. Data-driven modelling: some past experiences and new approaches. Journal of hydroinformatics, 10 (1), 3-22; Dibike, Y.B., Solomatine, D.P., 2001. River flow forecasting using artificial neural networks. Physics and Chemistry of the Earth Part B: Hydrology, Oceans and Atmosphere. Volume 26, Issue 1, Pages 1-7. c. The ebook "GFDRR. 2018. Machine Learning for Disaster Risk Management. Washington, DC: GFDRR.

[Figure]

License: Creative Commons Attribution CC BY 3.0. May be others are also included,

but I do believe a much wider bibliography is needed to support this work. In fact, the
first two are surely crucial supporting references, but are not adequately representing
and supporting the submitted research/perspective, considering that after 2001 and
2006 tremendous advancements and scientific production was developed not only for
ML in general, but within the topic of ML for floods specifically (see quick search on
SCOPUS below in Figure 1 and Figure 2).

SEE PDF Figure 1. All journal papers filtered by TITLE-ABS-KEYWORDS using search
criteria "Machine learning" AND "floods"

SEE PDF

Figure 2. All journal papers filtered by KEYWORDS using search criteria "Machine
learning" AND "floods"

2) Summary and review of state of the art ML.

I'd suggest authors to better introduce ad categorize major concepts, procedures and
tools of ML. In the introduction the reference to Bishop's book is then followed by few
specific examples (see also specific comments). I'd see here a flow chart or summary
table to improve the manuscript while addressing this general remark. I'm sure authors
can benefit and extrapolate the work already done and cited within the GFDRR book
on the topic. Additional scientific referencing can further strengthen this important part
of the manuscript where the reader will be guided with key concise definitions and
adequate referencing on state of the art ML for earth/geo/water science and flood risk
management in particular. Please see this general comment as a further extension of
general comment n.1, a surely expected contribution by authors to better support and
introduce the final findings/perspectives of this work.

3) Structure of the manuscript Sections 2, 3 and 4 should be merged into a Section
2 with subsections. I see the three components "exposure, hazard and impact" as a

unique core section with subsections related to descriptive versus predictive assessment models and related comments.

4) Scientific soundness of the "Perspectives" section

Section 5 seems to be a bit general. As requested to Invited perspectives my opinion is that authors miss to explicitly include in the paper sound arguments, facts, published research studies to support the conclusive remarks. Those remarks remain, in fact, general and simplicistic relying on few selected, yet relevant, references (mostly the GFDRR that is not even a research work). I understand this paper, as written in the Acknowledgement section, is the result of a "2 week long intensive collaboration during the Understanding Risk Field lab on urban flooding in Chiang Mai, but "Out of the context" and the intent is to share these ourcomes with the scientific audience, but the submitted manuscript seems not to capture the surely significant value of authors' knowledge and experience as well as the value of the Understanding Risk Field lab workshop discussion and reasoning.

Specific comments See attached commented PDF.

Please also note the supplement to this comment:
https://www.nat-hazards-earth-syst-sci-discuss.net/nhess-2019-341/nhess-2019-341-RC2-supplement.zip

---

## Author Comment (AC1) · 14 Feb 2020

Authors: We thank the reviewer for her/his time to read the paper and provide comments. We found the comments and suggestions helpful, and we have revised the paper accordingly. We think that the revised paper is a significant improved.

Reviewer 1: Terminology: as general assumption of our community (e. g. Merz et al., 2010, de Moel et al., 200), and following what authors write at P2 L43-46, flood impact is one of the three components of flood risk. Therefore, I would avoid to use the statement "flood risk and impact assessment", because flood impact is somehow included in flood risk. Please, go through the manuscript (include title) and correct these cases. Authors: Here we intended to make the distinction between flood risk (predictive and

probabilistic analysis conducted before an event) from descriptive flood impact assessment (observational and deterministic assessment conducted after an event). We have therefore added clarifying text in the introduction as follows: "We make the distinction between flood risk, as the probabilistic analysis of the potential (predictive) impacts of floods and flood impact assessment, as the post-event assessment of (descriptive) impact from an actual flood event."

Reviewer 1: Introduction: I would add some more detailed explanations of the different machine learning methods. Just some sentences, but it can help in order to have clear in mind, in the rest of the manuscript, what decision trees, neural network, etc. are. In addition, as general comments, I would add some sentences (and references) which state that machine learning methods can really improve estimations, in case of large datasets available: up to now, this concept is taken for granted, but citing some studies that demonstrate it could improve the manuscript, in my opinion. Authors: Thank you for this comment. We intended to highlight applicability and relevance of ML rather than details of specific methods. However, we have added some additional description of the methods and references, to give the reader a bit more context to the ML methods mentioned, and reference to learn more.

Reviewer 1: Structure: in order to be consistent with the definition of flood risk at P2 L43-49, I would suggest to analyse hazard (current Ch. 3) before exposure (current Ch. 2). Authors: We agree with the reviewer that the order of the chapters should correspond to the order in which the different components are introduced. We therefore changed the order in which we introduce the different components in the introduction.

Reviewer 1: Ch. 5: I would add, as an issue, the difficulty to use machine learning methods: they require a quite high degree of knowledges in order to really appreciate improvements in flood risk estimation and to avoid errors, that can easily be done by not-experts. Authors: We would like to thank the reviewer for this suggestion. We think the hype around machine learning can cause misuse and unwarranted trust by users in the models. We added an entire new paragraph to address this issue in chapter

5. Reviewer 1: P3 L89-90: Why traditional process models were not displaced by machine learning methods? Please explain, or refer to specific following Sections. Authors: That's an interesting question. This is implicitly already answered throughout the paper when we discuss future changes and obstacles for these changes to occur. However, we also added a sentence about this directly on page 3 so the reader doesn't have to wait for an answer.

Reviewer 1: P11 L352-353: I would suggest to remove the sentence. Authors: We considered removing this sentence but since this is the only appropriate place to reference to table 2 (see comment below), we decided to keep the sentence.

Reviewer 1: P12 L383-384: please add reference for this statement Authors: This sentence is a prediction/perspective that is then motivated in the rest of the paragraph. Within this motivation we added a few additional references now that would make this statement more convincing.

Reviewer 1: Table 2 is never cited in the text. Please correct Authors: We added a reference to this table

Reviewer 1: As suggestion, I list some recent papers I found, which use machine learning methods in the flood risk assessment, and can be useful to cite in order to strengthen some concepts: Authors: Thank you for these suggestions. These papers can help as references throughout the paper and have been included as additional motivation for existing text.

Please also note the supplement to this comment:
https://www.nat-hazards-earth-syst-sci-discuss.net/nhess-2019-341/nhess-2019-341-AC1-supplement.pdf

---

## Author Comment (AC2) · 14 Feb 2020

Response reviewer 2 Authors: We thank the reviewer for her/his time to read the paper and provide very detailed comments. We found the comments and suggestions helpful, and we have revised the paper accordingly. We think that the revised paper has been significantly improved.

Reviewer 2: Lack of adequate referencing Key references of this work are the following: a. Bishop, C. M. (2006), Pattern Recognition and Machine Learning, Springer, ISBN 978-0-387-31073-2 b. The two works: Solomatine, D.P., Ostfield, A. Data-driven modelling: some past experiences and new approaches. Journal of hydroinformatics, 10 (1), 3-22; Dibike, Y.B., Solomatine, D.P., 2001. River flow forecasting using artificial

neural networks. Physics and Chemistry of the Earth Part B: Hydrology, Oceans and Atmosphere. Volume 26, Issue 1, Pages 1-7. c. The ebook "GFDRR. 2018. Machine Learning for Disaster Risk Management. Washington, DC: GFDRR.

May be others are also included, but I do believe a much wider bibliography is needed to support this work. In fact, the first two are surely crucial supporting references, but are not adequately representing and supporting the submitted research/perspective, considering that after 2001 and 2006 tremendous advancements and scientific production was developed not only for ML in general, but within the topic of ML for floods specifically (see quick search on SCOPUS below in Figure 1 and Figure 2).

Authors: Thank you for this feedback. We have added 21 additional references to further support the paper. As such and while still not exhaustive, the list of 96 references included in our manuscript cover key contributions relevant to understand current state of the art in applications of machine learning methods for flood risk and impact analysis.

Reviewer 2: Summary and review of state of the art ML I'd suggest authors to better introduce ad categorize major concepts, procedures and tools of ML. In the introduction the reference to Bishop's book is then followed by few specific examples (see also specific comments). I'd see here a flow chart or summary table to improve the manuscript while addressing this general remark. I'm sure authors can benefit and extrapolate the work already done and cited within the GFDRR book on the topic. Additional scientific referencing can further strengthen this important part of the manuscript where the reader will be guided with key concise definitions and adequate referencing on state of the art ML for earth/geo/water science and flood risk management in particular. Please see this general comment as a further extension of general comment n.1, a surely expected contribution by authors to better support and introduce the final findings/perspectives of this work.

Authors: We described the state of the art of machine learning for flood risk assessments with numerous references within specific sections on exposure, hazard and impact. We choose this structure so we can, for each topic, transition easily from what is already being done to what could be done in the future, and directly discuss the difference between science and application. This gap between science and application causes a lot of grey area in the state-of-the-art description and makes it therefore necessary to do this in the main body of the paper. In a traditional research paper this information would indeed all be expected in the introduction, but this paper is a perspective on the latest scientific advances and how they can be extrapolated and be applied. The introduction therefore remains on a somewhat higher level of abstraction. We have further added a description of the paper structure up front in the in the introduction. As for the state of the art on machine learning techniques themselves, we deliberately focused on applications of these methods (i.e. what can you do with it) rather than theory (i.e. how do they work). For each method presented, we included references to papers and books that describe the specifics of the algorithms and underlying theory. Our target audience are flood risk management experts interested to understand the potential applications of these methods. We believe that covering theory of the ML methods would be beyond the scope of the paper, and detract readers from the main messages communicated. We did expand the description of the advances in machine learning to better highlight what is possible with the latest innovations that wasn't possible before (see also response to reviewer 1).

Reviewer 2: Structure of the manuscript Sections 2, 3 and 4 should be merged into a Section 2 with subsections. I see the three components "exposure, hazard and impact" as a unique core section with subsections related to descriptive versus predictive assessment models and related comments.

Authors: We merged sections 2, 3 and 4 into one section.

Reviewer 2: Scientific soundness of the "Perspectives" section Section 5 seems to be a bit general. As requested to Invited perspectives my opinion is that authors miss to explicitly include in the paper sound arguments, facts, published research studies to support the conclusive remarks. Those remarks remain, in fact, general and simplicistic

relying on few selected, yet relevant, references (mostly the GFDRR that is not even a research work).

Authors: Section 5 only contains general perspectives that apply to multiple topics and are seen throughout the paper or summaries of key points. Specific perspectives are covered in the exposure, hazard and impact sections and sometimes repeated here and classified by future likelihood. There is argumentation for the future likelihood. We have now added references to these earlier chapters in the perspectives chapter. In this way we now emphasize where the general comments are coming from and that they often arise from the specific chapters. We also changed the name of the chapter from "perspectives" to "general perspectives" so that we take away the expectation that only this chapter has perspectives.

Reviewer 2: I understand this paper, as written in the Acknowledgement section, is the result of a "2 week long intensive collaboration during the Understanding Risk Field lab on urban flooding in Chiang Mai, but "Out of the context" and the intent is to share these ourcomes with the scientific audience, but the submitted manuscript seems not to capture the surely significant value of authors' knowledge and experience as well as the value of the Understanding Risk Field lab workshop discussion and reasoning. Authors: We have re-written the acknowledgement section to better reflect the role of the event as a convening and facilitating platforms for multi-disciplinary experts, from which emerged much of this manuscript.

Reviewer 2: Line 16-20: I think the second sentence is redundant, merge into one sentence. Authors: We have modified this sentence to clarify that one refers to the past and the other to the future.

Reviewer 2: Line 27: "avoid repetitions just here rephrase" and line 29: "same comment as before" Authors: We removed some of the repetition.

Reviewer 2: Line 33: I'd expect a clear definition of machine learning as well a solid reference to this statement Authors: We repeated the reference from line 35 here now

because that also covers this statement. A more solid definition of machine learning is provided in line 69-70 .

Reviewer 2: Line 39: Again, as remarked before, I see redundancies. You are repeating that advantages vs limitations shall be considered. No references added here also makes the introuction weak. See general comment n.1

Authors: The second sentence was removed and the first has been integrated with the one prior.

Reviewer 2: Line 42: hazard, exposure and impact are key concepts of flood risk. Why mentioning here floods and society. And, again, why no references supporting key statements of the introduction? Authors: Exposure and impact are only relevant for floods when the relationship between floods and society is studied. The "interaction between floods and society" is a less technical term then just "flood risk". We added a reference to this key concept now. Reviewer 2: Line 51: I don't find a close link between this introductory sentence of ...application of floof risk and impact assessment, and the disaster managementy cycle

Authors: The disaster management cycle is a paradigm that includes everything that is done to fight the negative consequences of floods. Every flood risk and impact assessment should therefore have a benefit somewhere in this paradigm. The second line also refers directly to the first sentence already making the link clear, we strengthened that reference even more by repeating the word "different" before applications.

Reviewer 2: Line 57-58: you are here referring to recostruction of real events. I'd make this clear and cite relevant papers Authors: We are not referring to reconstructing hazard events. For a reconstruction you need to know what happened. We are talking about having a descriptive hazard (e.g. remote sensing based observation) and then developing a model to predict the impact. We rewrote this sentence to clarify it.

Reviewer 2: Line 63: there are two references by Wagenaar et al 2019 (make sure the

manuscript not accepted is removed from the list)

Authors: The first is already published. The second is under final review and will be published shortly. Since it is very relevant to the accompanying statement, we feel that it should be included as a reference, and should be made available to readers as relevant literature. Our understanding is that we should reference relevant literature, even if "unpublished" or "grey" literature.

Reviewer 2: Line 69: improve the introduction to machine learning, from definition to key concepts, procedures, tools etc

Authors: The paragraph starts with a definition, then clarifies this definition with an example. Procedures and tools go a bit unnecessarily deep for this paper see the response to the general comment "Summary and review of state-of-the-art ML". Reviewer 2: Line 73 a reference is needed here, otherwise it becomes quite unclear the reading Authors: We have added a reference.

Reviewer 2: Line 80 -85: I think this should be absolutely expanded, supported by more references. I advise the use of a summary table or flow chart. This paper can's comply with the title without a deep description of state of the art ML. See General comment n.2

Authors: See the response to general comment 2. The paragraph is expanded to make a clarify what the new methods can do compared to the old methods. However, we want to stick to the choice to not get into the topic of how the techniques work as it isn't necessary to answer the question in the title.

Reviewer 2: Line 129: reference?

Authors: The first part of the sentence is a conclusion of our literature research that is shown just before this sentence where we found no more advanced applications. The second part of the sentence is a perspective. This perspective is now supported with some additional reasoning.

Reviewer 2: Line 136: references?

Authors: This is about a categorization we defined for this paper. See introduction so it won't be possible/necessary to reference to that.

Reviewer 2: Line 168: there are tons of papers on the use of crowdsouced data for flood hazard modelling and mapping.

Authors: We referenced here to 4 recent papers about this and added a sentence about something that isn't done yet but could potentially be done in the future.

Reviewer 2: Line 207: this is not included in the reference list. The reference should be this https://arxiv.org/abs/1901.09583 but I don't find it proper for supporting this statement

Authors: That is indeed the correct reference, and thank you for pointing out that it is missing from the reference list. In the referenced paper, the authors suggest that hybrid ML-physical models will contribute to flood forecasting. Specifically, ML models 'responsible for calibration, error corrections, and perhaps additional processes that were not well modelled'. In this respect, we feel that using the paper to support our own argument (see below) is justified. "Hybrid component models assign machine learning a specific task in the modelling process that is either highly complex or not well understood. Examples of this include using machine learning for error correctors (see, for example, studies by Abrahart et al. 2007 and Google Research - Nevo et al. 2019)"

Reviewer 2: Line 208-215: See General comments n.1 and n.2 I suggest inserting more references and also more clear descriptions of the ML modelling approaches.

Authors: We already reference to 5 papers in this paragraph and all key ideas are already supported by at least 1 reference. We added a newer reference on the real-time control of flood defences and systems. We don't intend to get into more detail about it, this is just one of many applications in this paper and this level of detail is

appropriate for a high level understanding of the possibilities.

Reviewer 2: Line 220: all capital letters to define acronym

Authors: We changed this

Reviewer 2: Line 244: This may be generally true, but social networks are providing highe amount of geotagged images and videos. This is in contrast to what authors stated in the hazard assessment when citing crowdsource data.

Authors: This sentenced refers to labelled data that could be applied for training descriptive impact models (not for hazard assessment). Data from social networks is I expect only very rarely labelled with an object damage label that can be applied for training impact models. For descriptive hazard modelling the potential of social media data is already mentioned.

Reviewer 2: Line 248-253: This is quite superficial and not fully developed. What about drones, webcams and other sensors? Just to cite an example to trigger the discussion

Authors: We added a sentence that change detection might be applied to estimate damage from other angles then top view.

Reviewer 2: Line 281: check English

Authors: We will rewrite this sentence

Reviewer 2: Line 304: This entire section doesn not provide insightful comments and remarks on what's coming next. It is not, thus, respecting what was expected from the paper goal and title. The lack of proper introduction, definition and review of ML methods is probably affecting the robusteness of this section that doesn't go deeper in the discussion of perspectives of ML for flood risk management. See General comment n.3

Authors: See the response to general comment 3. In short the perspectives are included all over the paper and this section only summarizes general trends and classifies the likelihood of specific predictions earlier in the paper using argumentation. We added references to earlier section to better to make this link between this chapter and the rest of the paper clearer. Reviewer 2: Line 323: These 4 references are all missing from the reference list Authors: Thanks for pointing this out. We added them now.

Reviewer 2: Line 359: too general. Socia media will help what? exposure, hazard, impact or what?? You can't here just mention flood risk management Authors: We agree this is not well formulated. We rephrased this and referred to earlier chapters.

Please also note the supplement to this comment:
https://www.nat-hazards-earth-syst-sci-discuss.net/nhess-2019-341/nhess-2019-341-AC2-supplement.pdf

―――――――――――――――――――

---

## Referee Report (RR1)

**Review by Fernando Nardi (fernando.nardi@unistrapg.it)**

I am glad that authors positively considered the comments and remarks received in the first round of review. The manuscript was accordingly edited and it is significantly improved.

However, I still think there are some missing points that relate to ML and flood-related DRR that authors caught but didn't explore at full. In particular I'm sharing here two major comments that I'd like authors to check and eventually use for further improving the manuscript before publication. These two general comments relate to some aspects, of interest of ML for flood risk, that authors didn't consider or explore at full and in particular 1) New data, crowdsourced data and citizen science; and 2) human behavior and the value of ML for trans-disciplinary studies involving social, behavioral sciences and risk awareness, communication strategies

General comments/remarks

1) **New data, crowdsourced data and citizen science**. Authors cite "new data", but without providing a definition or a detailed conceptualization of what it is meant for new data in relation to the manuscript topic. Crowdsourced data and data/information from social network are also mentioned several times, but citizen science is never mentioned. There is an increasing trend, impact and production of citizen science project, social network based initiative for gathering, processing and modelling data of interest of flood risk management. The relationship and potential use of ML for empowering flood risk management by means of these new data is clear. I'd advise authors in this regard to check and cite these relevant works:

- Annis A. and Nardi F. (2019): Integrating VGI and 2D hydraulic models into a data assimilation framework for real time flood forecasting and mapping, Geo-spatial Information Science

- Mazzoleni, M., Verlaan, M., Alfonso, L., Monego, M., Norbiato, D., Ferri, M., & Solomatine, D. P. (2017). Can assimilation of crowdsourced data in hydrological modelling improve flood prediction?. Hydrology and Earth System Sciences, 21(2), 839-861.

- Assumpção, T. H., Popescu, I., Jonoski, A., & Solomatine, D. P. (2018). Citizen observations contributing to flood modelling: Opportunities and challenges. Hydrology and Earth System Sciences, 22(2), 1473-1489.

- Fohringer, J., Dransch, D., Kreibich, H., & Schröter, K. (2015). Social media as an information source for rapid flood inundation mapping. Natural Hazards and Earth System Sciences (NHESS), 15, 2725-2738.

2) **Human behavior and the value of ML for trans-disciplinary studies involving social, behavioral sciences and risk awareness, communication strategies**. An additional element that was never considered is the "human behavior" component that related to how citizens and in general the population behaves under flooding threats. The perception, the science literacy, and the way people respond before, during and after flooding events are all relevant factors that are missing in this work. This aspect goes beyond the development of more accurate or timelydata/information, either descriptive or predictive, and tackles a very important factor that is the way people act and react, the knowledge base and the general understanding of flood hazard. Moreover, communication and awareness are also not fully explored. I do believe ML will have a role in these components of the

Disaster Management Cycle. ML may also play a crucial role in the need of developing transdisciplinary studies integrating earth/hydrological sciences with social, behavioral and communication sciences

As a result, as invited perspective, deriving from outcomes of the june 2019 UR workshop, keeping in mind this manuscript provides a sort of an extended meeting report , I agree that the manuscript can be published as it is. Nevertheless, I'd be glad if authors could share their ideas, thoughts and applied research references related to the two above mentioned points and provide a final version of the manuscript for publication

**Specific comments**

See attached commented PDF with further specific comments and remarks

[revised manuscript text omitted]

---

## Author Response (AR2)

We would like to thank Dr. Fernando Nardi for his thoughtful comments and for going through the paper one more time. We went through his supplement and made technical corrections where he requested it.

We fully agree with the reviewer on his remarks about new data from crowd sourcing and social media. Both topics have been mentioned in the manuscript already but on the reviewers request we have mentioned them again in a sentence about new data. We also have included all the suggested references on these topics by the reviewer since they neatly fit into the paper.

The topic of communication and the role of Machine Learning in that is interesting but didn't come up during the Field Lab discussions or in the previous review round. We therefore decided not to add this topic to the manuscript but we added a sentence stating that this topic is outside the scope of the paper.